

# Ethanolic extract of Ya-nang (*Tiliacora triandra*) leaf powder induces apoptosis in cholangiocarcinoma cell lines via induction of hyperacetylation and inhibition of growth signaling

Arunta Samankul[1], Gulsiri Senawong[1], Prasan Swatsitang[1], Banchob Sripa[2], Chanokbhorn Phaosiri[3], Somdej Kanokmedhakul[3] and Thanaset Senawong[1]

[1] Department of Biochemistry, Faculty of Science, Khon Kaen University, Khon Kaen, Thailand
[2] Department of Pathology, Faculty of Medicine, Khon Kaen University, Khon Kaen, Thailand
[3] Department of Chemistry, Faculty of Science, Khon Kaen University, Khon Kaen, Thailand

## ABSTRACT

**Objective**. To develop alternative medicine for reducing undesired side effects of chemotherapy in CCA patients, the anticancer activity of *Tiliacora triandra* leaf powder ethanolic (TLPE) extract against cholangiocarcinoma cell lines was investigated.

**Methods**. Antiproliferation was studied using the MTT assay while apoptosis induction and cell cycle arrest were analyzed by flow cytometry. The levels of key proteins and phenolic acid content were analyzed by western blotting and reversed-phase HPLC, respectively.

**Results**. TLPE extract inhibited CCA cell growth in a dose- and time-dependent manner, with $IC_{50}$ values of $7.86 \pm 0.05$ µg/ml for KKU-M213B cells and $8.59 \pm 0.36$ µg/ml for KKU-100 cells at an exposure time of 72 h. TLPE extract inhibited the growth of CCA cell lines by inducing apoptosis of both cell lines and causing an increased population of KKU-100 cells at G0/G1 phase. TLPE extract up-regulated Ac-H3 but down-regulated p-ERK, p53, Bax, CDK4 and Bcl2 expressions in KKU-M213B cells. TLPE extract up-regulated Ac-H3, p21 and Bax but down-regulated p-ERK, p53, CDK4 and Bcl2 expressions in KKU-100 cells. Additionally, phenolic acids including *p*-hydroxybenzoic, vanillic, syringic, *p*-coumaric, ferulic and sinapinic acids were identified.

**Conclusion**. These results suggest the possibility of developing *T. triandra* leaf powder ethanolic extract as a chemotherapeutic or chemoprevention agent for cholangiocarcinoma.

Corresponding author
Thanaset Senawong,
sthanaset@kku.ac.th

## INTRODUCTION

Cholangiocarcinoma (CCA) is a bile duct cancer and the second most common primary liver neoplasm after hepatocellular carcinoma (*Tyson & El-Serag, 2011*; *Bergquist & von Seth, 2015*). The previous report shows that northeastern Thailand has the highest incidence

of CCA in the world (*Jongsuksuntigul & Imsomboon, 2003*). The infection of *Opisthorchis viverrini* has been classified as a risk factor for this disease (*Miwa et al., 2014*). CCAs are heterogeneous biliary epithelial tumors that are classified into intrahepatic (iCCA), perihilar (pCCA) and distal (dCCA) subtypes based on their anatomic location within the biliary tree. Especially in the recent decade, the incidence of iCCA has been increasing. Unfortunately, the 5-year survival rate for CCA remains less than 10% (*Bertuccio et al., 2019*; *Everhart & Ruhl, 2009*).

Most CCA patients are diagnosed in the late stages; thus, chemotherapy is very important for the treatment of CCA patients. Chemotherapy is also used as adjuvant chemotherapy in combination with surgery and may be used as neoadjuvant chemotherapy before surgery or radiation. The current chemotherapeutic drugs for cholangiocarcinoma are 5-fluorouracil, gemcitabine and cisplatin (*Valle et al., 2010*); however, drug resistance, toxicity and drug cost remain the significant limitations to the clinical use of these drugs. Therefore, herbal medicine is an alternative for CCA patients in developing countries, where medicinal herbs are being increasingly recognized as complementary treatments for cancer (*Aras et al., 2014*; *Shukla, 2007*).

Ya-nang (*Tiliacora triandra*) is classified as a member of the Menispermaceae family. It is a member of a genus of flowering plants native to Southeast Asia, especially in northeast Thailand and the Lao People's Democratic Republic (LPDR) (*Singthong, Ningsanond & Cui, 2009*). In Thailand and Laos, the leaves of *T. triandra* are extracted with water by rubbing them with hands to produce a product called Nam Ya-Nang which is used as a cooking ingredient for bamboo shoot soup and bamboo shoot salad. In addition, *T. triandra* is used as traditional medicine to cure illnesses such as fever and malaria and it is also used as an anti-pyretic drug, detoxication agent, anti-inflammation agent, anti-cancer agent, anti-bacterial agent, immune modulator, and anti-oxidant (*Rattana, Padungkit & Cushnie, 2010*). Phytochemicals of *T. triandra* such as tiliacorinine, 2′-nor-tiliacorinine, tiliacorine and 13′-bromo-tiliacorinine were isolated from the root, which contained many different types of four bisbenzylisoquinoline alkaloids (*Sureram et al., 2012*). Moreover, the compound oxoanolobine was isolated from the *T. triandra* leaves (*Rattana et al., 2016*). *T. triandra* leaves exhibited high phenolic content (*Singthong et al., 2014*; *Phunchago, Wattanathorn & Chaisiwamongkol, 2015*), flavonoid (*Rattana, Padungkit & Cushnie, 2010*), polysaccharide gum, which were identified by FTIR detection as xylan (*Singthong, Ningsanond & Cui, 2009*) and fatty acids including phytol, 1-cyclohexenylacetic acid, oleamide, oleic acid, palmitic acid, neophytadiene 5-hydroxymethyl-2-furancarboxaldehyde, 2,6-dimethyl-3 (methoxymethyl)-benzoquinone (*Chaveerach et al., 2016*), hexadecanoic, octadecanoic and (Z)-6 octadecenoic acids (*Kaewpiboon et al., 2014*). In this study, phenolic acids such as *p*-hydroxybenzoic, vanillic, syringic, *p*-coumaric, ferulic and sinapinic acids were identified from the *T. triandra* leaves. The ethanolic extract of *T. triandra* leaf, which contains these phenolic acids was demonstrated to possess anticancer activity against cholangiocarcinoma cell lines.

## MATERIALS & METHODS

### Chemicals

*Tiliacora triandra* leaf powder was purchased from the Rak Samunpai group, Bangkok, Thailand (https://www.thaiherbbiz.com). Camptothecin, Annexin V-FITC, propidium iodide (PI) and all primary and secondary antibodies were obtained as previously described (*Saenglee et al., 2016*; *Saenglee et al., 2018*).

### Cell lines and culture conditions

Well-differentiated (KKU-M213B) and poorly differentiated and drug-resistant (KKU-100) adenocarcinoma cell lines were kindly provided by Dr. Banchob Sripa (Khon Kaen University, Khon Kaen, Thailand). The non-cancer cell line (H69; human bile duct epithelial cell) was obtained from Dr. Douglas Jefferson (Tufts University, Boston, MA, USA). Cholangiocarcinoma cells and human bile duct epithelial cells were cultured in suitable conditions as previously described (*Saenglee et al., 2018*).

### Sample preparation

*T. triandra* leaf powder ethanolic (TLPE) extract was prepared for testing antiproliferation, induction of cell cycle arrest and apoptosis, regulation of protein expression levels and total phenolic content. A total of 20 g of *T. triandra* leaf powder was added in 200 ml of absolute ethanol and stirred for 48 h. The suspension was centrifuged at 6, 150× g (Centrifuge, Hitachi, R20A2) for 15 min. The supernatant was collected and filtered with Whatman No. 4 filter paper. The supernatant was evaporated to 2 ml using a rotary evaporator and finally to dryness under a gentle stream of nitrogen. The ethanolic extract was kept at −20 °C until used.

### Cell viability

The MTT (3-(4,5-dimethylthiazol-2-yl)-2,5-diphenyltetrazolium bromide) assay was performed to assess cell viability as described in a previous study (*Saenglee et al., 2016*). Briefly, KKU-M213B, KKU-100 and H69 cells were seeded in each well of 96-well plates ($8 \times 10^3$ cells/well) and incubated in $CO_2$ incubator for 24 h. Then, various concentrations of TLPE extract and solvent control (1% of absolute ethanol) were added in each well and incubated for 24, 48 and 72 h. Then, the culture medium was replaced with the medium containing 1.2 mM of MTT (100 μl/well) and incubated for 2–4 h. The purple formazan crystals were dissolved in DMSO. The absorbance at 550 nm and optical density at 655 nm (the reference wavelength) were measured using a microplate reader (Bio-Rad Laboratories, Hercules, CA, USA). The percentage of cell viability of each well was calculated using the equation:

$$\%\text{Cell viability} = [\text{Sample Abs}(Abs550 - O.D.655)/\text{Control Abs}(Abs550 - O.D.655)] \times 100.$$

### Analysis of cell cycle phase distribution by flow cytometry

Cell cycle progression was analyzed using propidium iodide staining and flow cytometry as described in a previous study (*Saenglee et al., 2018*). In brief, KKU-M213B, KKU-100

and H-69 cells were seeded at density of $1 \times 10^6$ cells/5.5 cm dish for 24 h. The cells were treated with TLPE extract (31.25, 62.5 and 125 $\mu$g/ml) and absolute ethanol as a solvent control for 24 h. The treated cells were harvested by trypsinization, centrifuged at 1,754 $\times$ g for 3 min and resuspended in ice-cold 1X PBS with 0.1% glucose. The cells were added with 70% ethanol, incubated at 4 °C for 1 h and centrifuged at 1,754 $\times$ g for 3 min. Cells were resuspended in ice-cold 1X PBS and centrifuged at 1,754 $\times$ g for 3 min. Then, the cells were added with 10 mg/ml RNase A and incubated at 37 °C. The cells were stained with 5.0 $\mu$l of PI and incubated in the dark at room temperature for 45 min. Cell cycle phases were analyzed by flow cytometry (Becton Dickinson, San Jose, CA, USA) (The service was provided by the Research Instrument Center, Khon Kaen University, Thailand).

## Analysis of apoptosis by flow cytometry

Apoptosis induction was evaluated by flow cytometry using Annexin V-FITC and PI staining as previously described (*Saenglee et al., 2018*). In brief, KKU-M213B, KKU-100 and H69 cell lines were cultured in a 5.5-cm dish at a density of $1 \times 10^6$ cells/dish for 24 h and treated as described in the analysis of cell cycle phase distribution. Then, cells were harvested and the cell pellet was rinsed twice with ice-cold PBS. The cell pellet was then resuspended in the Annexin-binding buffer (100 $\mu$L) and incubated in the dark with Annexin V-FITC and PI for 15 min at room temperature. Apoptotic cells were analyzed using a BD FACSCantoII Flow Cytometer.

## Western blot analysis

Cells were cultured and treated as described in the analysis of cell cycle phase distribution and apoptosis. The protein levels were detected by western blot as described in a previous study (*Saenglee et al., 2018*). Briefly, the harvested cells were lysed with RIPA lysis buffer. Total protein samples (30–80 $\mu$g) were loaded and separated through 10%–12.5% SDS-PAGE (15 mA for 1.30 h) and then transferred to a polyvinylidene fluoride (PVDF) membrane. After blocking with 5% skim milk for 1 h, the blots were incubated overnight with specific primary antibodies. The following antibodies were used: anti-Bcl2, anti-CDK4, anti-p21, anti-p53, anti-Bax, anti-acetylated H3, anti-pERK1/2 and anti-ERK1/2. The membrane was subsequently incubated with an HRP-conjugated secondary antibody for 2 h at room temperature. The Clarity™ Western ECL substrate (Bio-Rad, CA, USA) was used to develop the protein bands. The X-ray film (GE Healthcare Bio-Sciences, USA) was exposed to the immunoreactive bands.

## Phenolic acid extraction

A total of 20 g of *T. triandra* leaf powder was macerated in absolute ethanol at a ratio of 1:10 (20 g *T. triandra* leaf powder: 200 mL absolute ethanol). The mixture was stirred for 48 h at room temperature in the dark, centrifuged thereafter at 6,150$\times$ g for 15 min and filtered through Whatman grade No. 4 filter paper. The supernatant was then processed as previously described (*Woranam et al., 2020*) to obtain the phenolic acid extract. The phenolic acid extract was dissolved in 50% methanol, filtered through a 0.2 $\mu$m syringe filter, and analyzed for the type of phenolic acids using reversed-phase HPLC.

## HPLC analysis

To identify individual phenolic acids in the extract, reversed phase HPLC was used, and the identification was based on matching the spectrum and retention times of phenolic acid standards as described previously (*Khaopha et al., 2012*). Individual phenolic acids in TLPE extract were analyzed quantitatively by comparison with a standard curve and using *m*-hydroxybenzaldehyde as the internal standard.

## Total phenolic content

Total phenolic content was analyzed as previously described (*Khaopha et al., 2012*) based on the reduction of the Folin-Ciocalteu reagent (phosphomolybdic and phosphotungstic acids) by phenolic groups of phenolic compounds. Briefly, TLPE extract was mixed with Folin-Ciocalteu reagent and 20% $Na_2CO_3$ and adjusted the final volume to 200 µl by adding distilled water. The mixture was incubated at 50 °C for 2 h and then the absorbance was measured at 750 nm using a spectrophotometer. Total phenolic content was calculated and reported as mg gallic acid equivalent (GAE) per gram dry weight.

## Statistical analysis

Data were expressed as the mean ± standard deviation (SD) from three independent experiments. The significant differences between solvent control and treatments were analyzed as previously described (*Saenglee et al., 2018*).

# RESULTS

## Antiproliferative activity of TLPE extract against CCA cell lines

Anti-proliferative activity of TLPE extract against cholangiocarcinoma cell lines (KKU-100 and KKU-M213B cells) and non-tumorigenic biliary epithelial cells (H69 cells) was investigated using the MTT assay. The results were shown as percentages of cell viability and $IC_{50}$ values. TLPE extract had a dose- and time-dependent anti-proliferative effect against KKU-M213B, KKU-100 and H69 cells (Fig. 1). KKU-M213B cell line was the most sensitive to the TLPE extract. TLPE extract significantly inhibited the proliferation of KKU-M213B cells with $IC_{50}$ values of 90.50 ± 9.62, 9.09 ± 0.15 and 7.86 ± 0.05 µg/ml at exposure times of 24, 48 and 72 h, respectively (Figs. 1A–1B). TLPE extract significantly inhibited the proliferation of KKU-100 cells with $IC_{50}$ values of 115.32 ± 9.53, 13.67 ± 1.44 and 8.59 ± 0.36 µg/ml at exposure times of 24, 48 and 72 h, respectively (Figs. 1C–1D). In addition, the cytotoxicity of TLPE extract in non-tumorigenic biliary epithelial cells (H69 cells) was investigated in comparison with the cancer cell lines. TLPE extract exhibited less toxicity to non-cancer H69 cells with $IC_{50}$ values of 143.39 ± 8.52, 14.75 ± 0.60 and 11.95 ± 0.60 µg/ml at exposure times of 24, 48 and 72 h, respectively (Figs. 1E–1F).

## TLPE extract induces cell cycle arrest in CCA cell lines

In this study, the cholangiocarcinoma cell lines (KKU-M213B, KKU-100) and non-tumorigenic biliary epithelial cells (H69) were treated with TLPE extract at 31.25, 62.50 and 125.00 µg/ml for 24 h. KKU-M213B cells were treated with TLPE extract at 31.25, 62.50 and 125.00 µg/ml for 24 h but did not show accumulation of cells at any cell cycle phase.

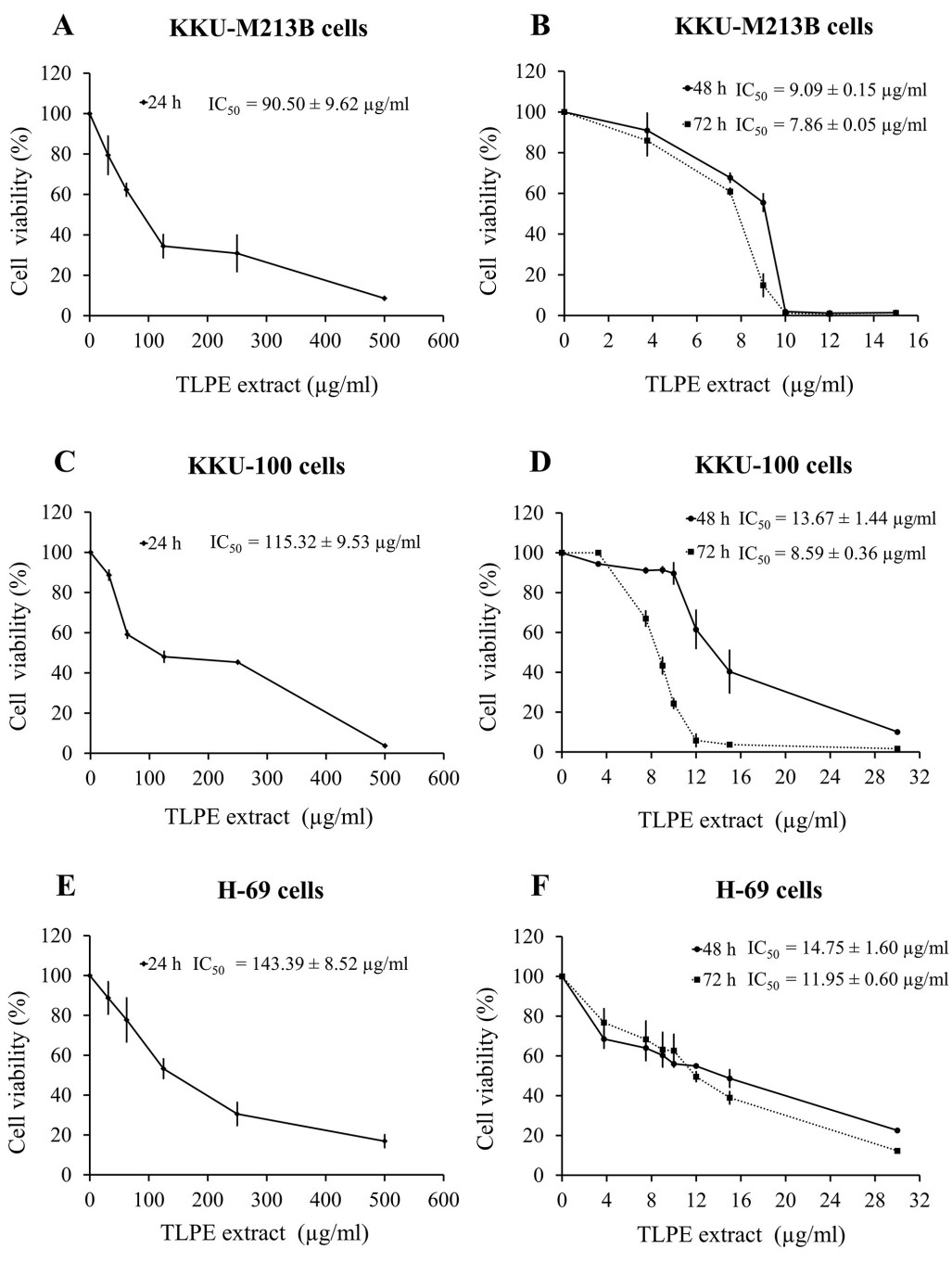

**Figure 1** **The effect of TLPE extract on the proliferation of CCA cells.** KKU-M213B (A, B), KKU-100 (C, D) and H69 (D, E) cells were treated with TLPE extract of 31.25–500 μg/ml for 24 h and 3.75–30 μg/ml for 48 and 72 h. Antiproliferative activity was determined using MTT assay according to manufacturer protocol. Cell viabilities are expressed as percentages of the solvent control, which was defined as 100% cell viability. The IC$_{50}$ values are indicated as the means ± S.D. from three independent experiments.

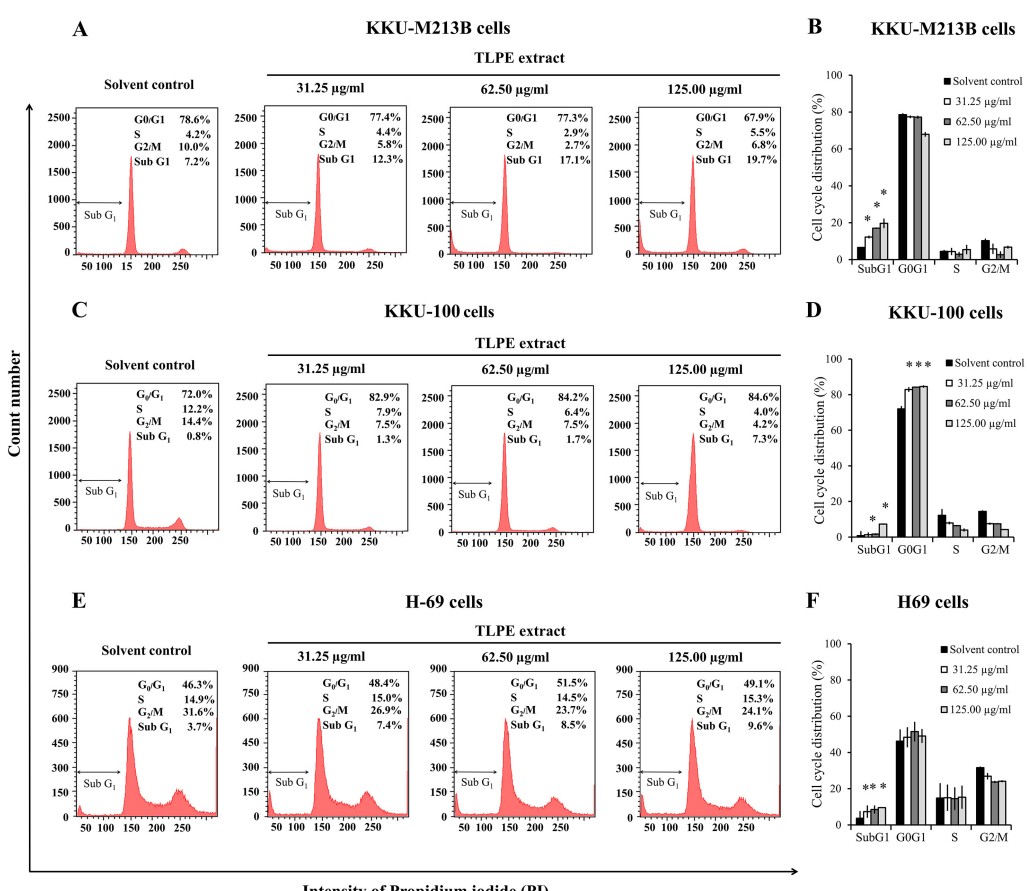

**Figure 2** **Histograms display the DNA content distribution of CCA cells.** KKU-M213B (A), KKU-100 (C) and H69 (E) cell cycle profiles after treated with TLPE extract. Cells treated with 1% EtOH (v/v), 31.25, 62.50 and 125.00 µg/ml of TLPE extract were stained with propidium iodide (PI) and analyzed by flow cytometry. Bar graphs represent the summarized data on cell cycle distribution of KKU-M213B (B), KKU-100 (D) and H69 (F) cells. The percentages of cells are presented as means ± S.D. from two independent experiments performed in duplicate. The asterisk "*" indicates a significant increase when compared with solvent control treatments ($p < 0.05$).

However, a significant increase in the cell population in the SubG1 phase was observed up to 12.3%, 17.1% and 19.7%, respectively (Figs. 2A–2B). KKU-100 cells treated with TLPE extract at 31.25, 62.50 and 125.00 µg/ml for 24 h induced cell cycle arrest at the G0/G1 phase (Figs. 2C–2D). The cell populations in the G0/G1 phase were significantly increased up to 82.9%, 84.2% and 84.6% when treated with TLPE extract at 31.25, 62.50 and 125.00 µg/ml, respectively. In addition, a significant increase in the cell population in the SubG1 phase was observed up to 1.3%, 1.7% and 7.3%, respectively (Figs. 2C–2D). H69 cells treated with TLPE extract at 31.25, 62.50 and 125.00 µg/ml for 24 h showed no accumulation of cells at any cell cycle phase (Figs. 2E–2F). However, a significant increase of the SubG1 population in H69 cells was observed up to 7.4%, 8.5% and 9.6%, respectively (Figs. 2E–2F).

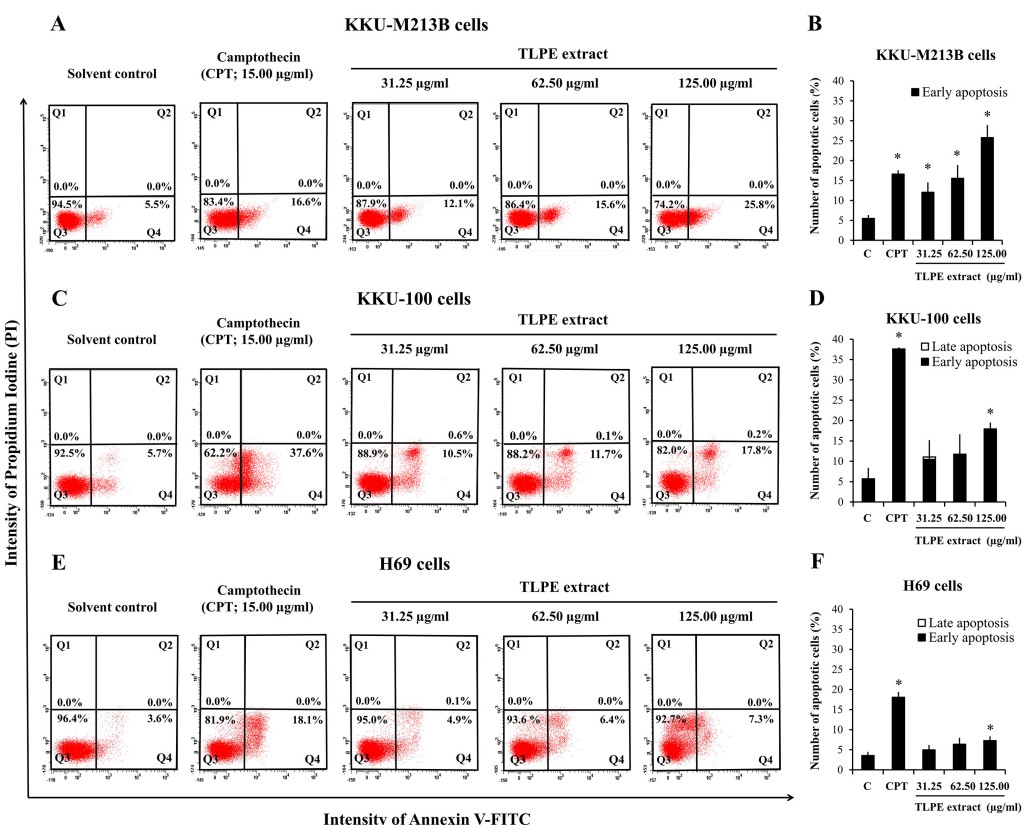

**Figure 3 Apoptotic death analysis by flow cytometry.** Representative dot plots display the apoptotic death of the cells treated with the indicated concentrations in KKU-M213B (A), KKU-100 (C) and H69 (E) cells treated with TLPE extract. After 24 h-treatment, cells labeled with Annexin V-FITC and Propidium iodide (PI) were analyzed by flow cytometry to determine the percentage of cells displaying an increase in early (panels Q4) and late (panels Q2) apoptosis. The cells treated with camptothecin (15 mg/mL) and absolute ethanol (1%; v/v) were used as positive and negative controls, respectively. Bar graphs show the summarized data on KKU-M213B (B), KKU-100 (D) and H69 (F) cells from two independent experiments performed in duplicate. Graphs represent the percentages of early and late apoptotic cells with means ± S.D. The symbol "*" indicates a significant difference ($p < 0.05$).

## TLPE extract induces apoptosis in CCA cell lines

Apoptosis induction was determined by Annexin V-FITC and propidium iodide (PI) staining. The stained cells were analyzed by flow cytometry. KKU-M213B cells treated with TLPE extract at 31.25, 62.50 and 125.00 µg/ml exhibited the increased percentages of early apoptotic cells, which were accounted for at 12.1%, 15.6% and 25.8%, respectively (Figs. 3A–3B). TLPE extract stimulated apoptosis induction in KKU-M213B cells in a dose-dependent manner for 24 h. In KKU-100 cells, early apoptotic cell populations were significantly increased to 10.5%, 11.7% and 17.8%, respectively (Figs. 3C–3D). The late apoptotic cell populations were 0.6%, 0.1% and 0.2%, respectively. In the non-cancer H69 cells, early apoptotic cell populations were slightly increased to 4.9%, 6.4% and 7.3% (Figs. 3E–3F), respectively.

**Table 1  Phenolic acid compositions of _T. triandra_ ethanolic extract.**

| Sample | Phenolic acids (mg/g of dry extract)[a] | | | | | |
|---|---|---|---|---|---|---|
| | _p_-Hydroxybenzoic acid | Vanillic acid | Syringic acid | _p_-Coumaric acid | Ferulic acid | Sinapinic acid |
| TLPE extract | $2.36 \pm 0.84$ | $2.56 \pm 0.13$ | $2.22 \pm 0.10$ | $4.16 \pm 0.11$ | $3.26 \pm 0.16$ | $6.16 \pm 0.31$ |

Notes.
[a]Results are expressed as means $\pm$ SD of three determinations.

### TLPE extract modulates expression of proteins involved in apoptosis and cell cycle arrest in CCA cell lines

The above results showed that TLPE extract induced apoptosis in both KKU-M213B (Figs. 3A–3B) and KKU-100 (Figs. 3C–3D) cells, respectively. Moreover, TLPE extract induced cell cycle arrest at G0/G1 in KKU-100 (Figs. 2C–2D) cells. To understand the molecular mechanisms by which TLPE extract induces apoptosis and cell cycle arrest in CCA cell lines, the expression levels of Bcl2 (anti-apoptotic protein), Bax (pro-apoptotic protein), p53 (tumor suppressor protein) and cell cycle-related proteins (p21; inhibitor of cyclin dependent kinases, CDK4; the progression through the G1 phase) were examined. TLPE extract decreased the level of p53 in KKU-M213B and KKU-100 cells when compared to the control (solvent control) (Fig. 4). TLPE extract also caused a decrease in the level of the anti-apoptotic protein Bcl2 in both cancer cell lines when tested at higher concentrations (Figs. 4A–4B, 4D–4E). The amount of the pro-apoptotic protein Bax was decreased in KKU-M213B cells but increased in KKU-100 cells. The relative ratio of Bax/Bcl2 was significantly increased at the high concentration treatments in both KKU-M313B and KKU-100 cells (Figs. 4C–4F).

TLPE extract caused a decrease in a protein level of CDK4 in both KKU-M213B (Figs. 4A–4B) and KKU-100 cells (Figs. 4D–4E) but did not affect the amount of p21 in KKU-M213B cells (Figs. 4A–4B). In KKU-100 cells, the protein level of p21 was increased, which was related to the induction of the cell cycle arrest at G1 phase in KKU-100 cells (Figs. 2C–2D). Ethanol extracts from most parts of the plant often contain _p_-coumaric acid and several phenolic acids that act as histone deacetylase inhibitors (HDAC inhibitors) (_Saenglee et al., 2016_). In this study, _p_-coumaric and sinapinic acids were the most predominant phenolic acids that were identified in TLPE extract (Fig. 5 and Table 1). Therefore, we tested whether TLPE extract could inhibit the activity of histone deacetylases in CCA cells. As shown in Fig. 4, the increased acetylated histone H3 protein was observed both in KKU-M213B and KKU-100 cells, indicating that TLPE extract possessed HDAC inhibitory activity.

In addition, the p-ERK1/2 levels were significantly reduced in both CCA cell lines at the concentrations tested (Figs. 4A–4B, 4D–4E), indicating that inhibition of the ERK1/2 signaling pathway may be implicated in TLPE extract-induced apoptosis in cholangiocarcinoma cells.

### Total phenolic content and type of phenolic acids in TLPE extract

The total phenolic content in TLPE extract was $47.67 \pm 6.77$ mg gallic acid equivalent/g of dry extract. Figure 5 shows typical chromatograms of phenolic acid standards (Figs. 5A) and the TLPE extract (5B). Six phenolic acids including _p_-hydroxybenzoic,

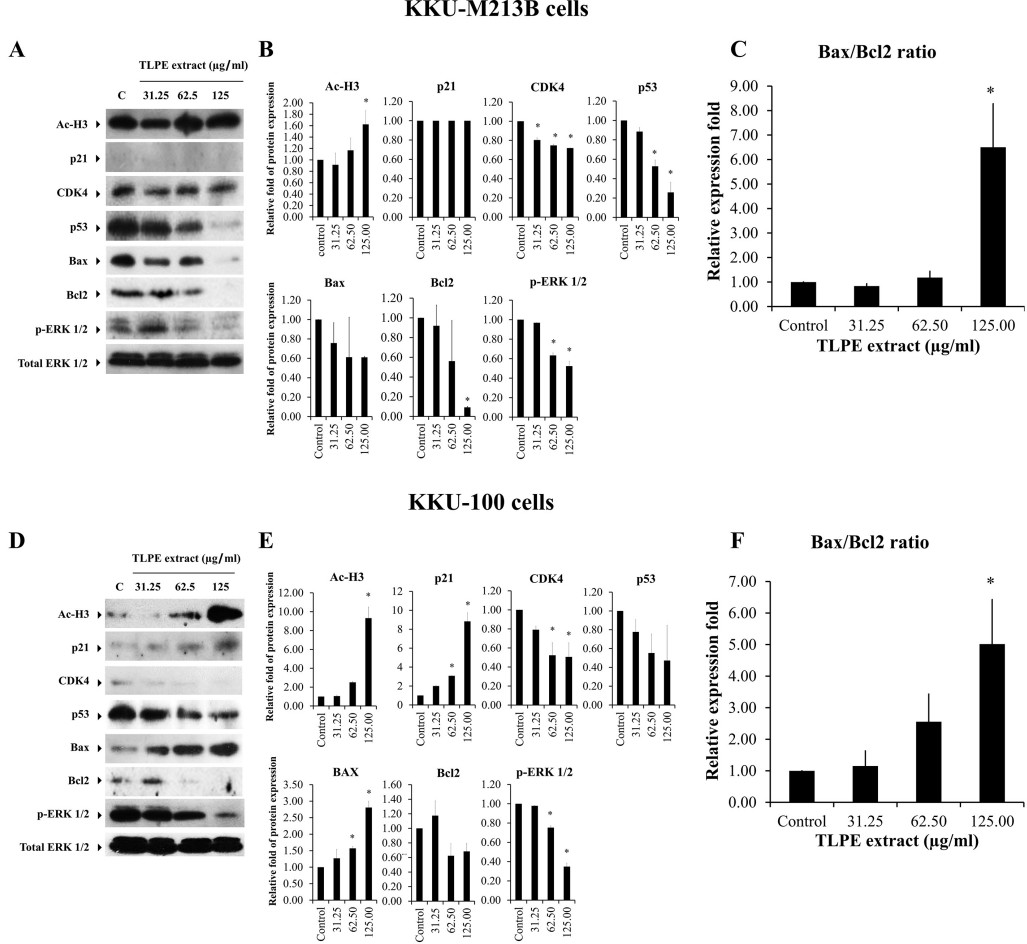

**Figure 4** **The effect of TLPE extract on cell cycle and apoptosis regulatory proteins in CCA cells.** KKU-M213B (A) and KKU-100 (D) cells were treated with absolute ethanol as control and TLPE extract at 31.25, 62.5 and 125 µg/ml for 24 h. After 24 h, the cells were harvested and proteins were extracted and separated on SDS-PAGE. Expression of p53, p21, CDK4, Bax, Ac-H3, Bcl2 and p-ERK were detected by immunoblotting. Total ERK was used as a loading control in western blot analysis. The intensity of protein bands was quantitated by densitometric analysis. The bar graphs represent the relative fold of protein expression in KKU-M213B (B) and KKU-100 (E) cells compared with internal control. The relative ratio of Bax/Bcl2 was calculated and found to be significant for the high concentration treatments in both KKU-M213B (C) and KKU-100 (F) cells.

vanillic, syringic, *p*- coumaric, ferulic and sinapinic acids were identified in TLPE extract with *p*-coumaric and sinapinic acids being the most pre-dominant (Table 1).

# DISCUSSION

The MTT results revealed that TLPE extract inhibited proliferation of both CCA cell lines in a dose- and time-dependent manner. In addition, non-tumorigenic biliary epithelial (H69) cells exhibited the lowest sensitivity to TLPE extract, indicating somewhat greater specificity to CCA cell lines of the extract. Similarly, *T. triandra* leaf powder methanolic extract has been shown to inhibit the growth of hepatocellular carcinoma cells (IC$_{50}$ = 0.41

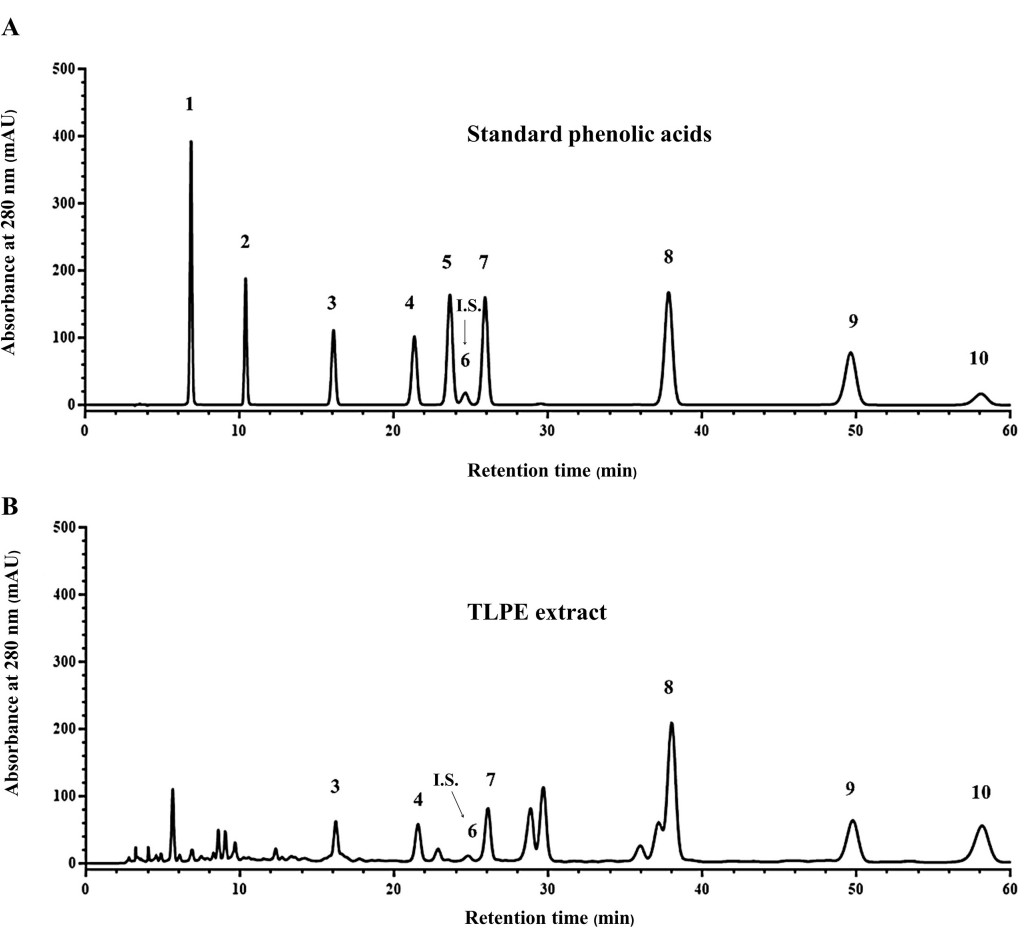

**Figure 5 HPLC chromatograms of the phenolic acids.** Standard phenolic acids (A) and TLPE extract (B): 1, gallic acid; 2, protocatechuic acid; 3, *p*-hydroxybenzoic acid; 4, vanillic acid; 5, caffeic acid; 6, *m*-hydroxybenzaldehyde; 7, syringic acid; 8, *p*-coumaric acid; 9, ferulic acid and 10, sinapinic acid.

mg/ml) but is less toxic to white blood cells (Lymphocytes) ($IC_{50}$ = 10 mg/mL) (*Chaveerach et al., 2016*). *T. triandra* leaf was previously extracted by various solvents (petroleum ether, dichloromethane, ethyl acetate, methanol and water) and was also investigated for its ability to inhibit the growth of oral (KB), lungs (NCI-H187) and breast (MCF7) cancer cells (*Rattana et al., 2016*). Methanol and water extracts of *T. triandra* leaf powder had the potential for growth inhibition of NCI-H187, KB and NCI-H187 cells ($IC_{50}$ <50 μg/ml). In addition, *T. triandra* leaf powder ether, dichloromethane and ethyl acetate extracts inhibited the growth of many cancer cells ($IC_{50}$ >50 μg/ml) (*Rattana et al., 2016*).

Recently, tiliacorinine was isolated from the root and stem of *T. triandra*, inhibiting proliferation of cholangiocarcinoma cell lines (KKU-055, KKU-213, KKU-214 and KKU-100) with $IC_{50}$ values of 4.50−7.00 μM (*Janeklang et al., 2014*). This evidence showed that the anticancer effect of *T. triandra* leaf might be related to the solvent used for extraction and the type of cancer cells. However, TLPE extract has not been studied previously on cholangiocarcinoma cell lines. The present study demonstrated that TLPE extract induced

apoptosis in two cholangiocarcinoma cell lines (KKU-M213B and KKU-100 cells). The toxicity of TLPE extract in non-tumorigenic biliary epithelial cells (H69 cells) was also investigated and less toxicity was found in this cell line compared to CCA cells. In the previous study, tiliacorinine isolated from *T. triandra*, induced apoptosis and caspase activity in cholangiocarcinoma (KKU-M213B, KKU-100) cell lines (*Janeklang et al., 2014*). In this study, natural HDAC inhibitors including *p*-coumaric, ferulic and sinapinic acids were identified in TLPE extract (Fig. 5 and Table 1). In addition, western blot results demonstrated that TLPE extract caused an increase in acetylated histone H3 protein content in both KKU-M213B and KKU-100 cells (Figs. 4A–4B, 4D–4E). Similarly, the compounds with HDAC inhibitory activity caused hyperacetylation with a more relaxed chromatin structure (*Fang, 2005*; *Mork, Faller & Spanjaard, 2005*).

The level of p53 protein was decreased in both types of cancer cells, indicating that the mechanism of stimulating apoptosis in cholangiocarcinoma cells may be p53-independent (*Hiebert et al., 1995*). Furthermore, the level of Bcl2 was decreased when the cells were treated with TLPE extract (Fig. 4). The reduction of the anti-apoptotic protein Bcl2, causes a release of cytochrome c from the mitochondrial membrane leading to the activation of apoptosis in the intrinsic pathway (*Nahle et al., 2002*). TLPE extract could induce cell cycle arrest at the G0/G1 phase in KKU-100 cells. TLPE extract caused the increased p21 protein levels in KKU-100 cells. p21 is an inhibitor of the cell cycle that can inhibit CDK4 activity causing the cell cycle arrest. However, it was found that TLPE extract caused a decrease in CDK4 protein levels in both KKU-M213B and KKU-100 cells (Figs. 4A–4B, 4D–4E). In KKU-100 cells, CDK4 protein levels were decreased which may be related to the cell cycle arrest at G1 phase. In KKU-M213B cells, the CDK4 protein level was slightly reduced, but there was no cell cycle arrest in the G1 phase. Moreover, the level of p21 protein was not increased in KKU-M213B cells. It has been reported that p21 down-regulation inhibits apoptosis by binding with procaspase 3 and blocking the protease cleavage site (*Suzuki et al., 1999*). In addition, we hypothesized that TLPE extract may cause an increase in the level of pro-apoptotic proteins such as Bax protein. In this study, TLPE extract caused a decrease in the level of Bax protein in KKU-M213B cells but an increase in KKU-100 cells (Figs. 4A–4B, 4D–4E). However, the relative ratio of Bax/Bcl2 was significantly increased in both KKU-M213B and KKU-100 cells (Figs. 4C, 4F), which was corresponding to an increased apoptosis at a higher concentration of TLPE extract treatment (Figs. 3A–3B). Due to the reduced Bax protein levels in KKU-M213B cells, it has been reported that Bax down-regulation involves p53 mutations (*Beerheide et al., 2000*).

## CONCLUSIONS

TLPE extract inhibited the growth of CCA cell lines. KKU-M213B cells were more sensitive to TLPE extract than KKU-100 cells. In addition, TLPE extract was less toxic to non-tumorigenic biliary epithelial cells (H69) than to both cancer cell lines. TLPE extract inhibited both human cholangiocarcinoma cell lines via apoptosis induction. Induction of cell cycle arrest was observed at the G0/G1 phase only in KKU-100 cells. Apoptosis and cell cycle arrest inductions were confirmed by the levels of protein expression involved.

Moreover, the analysis of the type of phenolic acids in TLPE extract revealed six phenolic acids: (1) *p*-hydroxybenzoic acid, (2) vanillic acid, (3) syringic acid, (4) *p*-coumaric acid, (5) ferulic acid and (6) sinapinic acid. Accordingly, TLPE extract has potential for development as an alternative medicine for human cholangiocarcinoma treatment.

## ACKNOWLEDGEMENTS

We would like to thank Associate Professor Dr. Albert J. Ketterman, Mahidol University, Thailand, for English proofreading.

### Funding

This research was funded by the National Science Research and Innovation Fund through Khon Kaen University (Re-submit-2563). Arunta Samankul was financially supported by Thailand and Research Professional Development Project Under the Science Achievement Scholarship of Thailand (SAST). There was no additional external funding received for this study. The funders had no role in study design, data collection and analysis, decision to publish, or preparation of the manuscript.

### Grant Disclosures

The following grant information was disclosed by the authors:
National Science Research and Innovation Fund through Khon Kaen University: Re-submit-2563.
Thailand and Research Professional Development Project Under the Science Achievement Scholarship of Thailand (SAST).

### Competing Interests

The authors declare there are no competing interests.

### Author Contributions

- Arunta Samankul performed the experiments, analyzed the data, prepared figures and/or tables, and approved the final draft.
- Gulsiri Senawong conceived and designed the experiments, authored or reviewed drafts of the article, and approved the final draft.
- Prasan Swatsitang analyzed the data, authored or reviewed drafts of the article, and approved the final draft.
- Banchob Sripa analyzed the data, authored or reviewed drafts of the article, and approved the final draft.
- Chanokbhorn Phaosiri performed the experiments, analyzed the data, prepared figures and/or tables, authored or reviewed drafts of the article, and approved the final draft.
- Somdej Kanokmedhakul analyzed the data, authored or reviewed drafts of the article, and approved the final draft.
- Thanaset Senawong conceived and designed the experiments, analyzed the data, prepared figures and/or tables, and approved the final draft.

## Data Availability

The original images for blots and raw data are available in the Supplementary Files.

## Supplemental Information

Supplemental information for this article can be found online at http://dx.doi.org/10.7717/peerj.14518#supplemental-information.

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
