# Peer review of "Ethanolic extract of Ya-nang (Tiliacora triandra) leaf powder induces apoptosis in cholangiocarcinoma cell lines via induction of hyperacetylation and inhibition of growth signaling"

_PeerJ, doi:10.7717/peerj.14518_

## Round 0.1 · original submission · Major Revisions

Manuscript does not fulfill the standards established for the journal to be considered for publication in its current form. I agree with the reviewers that manuscript requires substantial revision to support the conclusion and improve the quality of the publication. Please see the detailed comments made by the reviewers.

Moreover, thorough English editing is required. Please revise the manuscript taking help from a colleague who is proficient in English and familiar with the subject matter, who can review your manuscript, or contact a professional editing service to review your manuscript. Revise and resubmit accordingly.

Reviewer 1 ·

Basic reporting

no comment

Experimental design

no comment

Validity of the findings

To ensure the validity of the results of this research, the following additional data are needed:
1. Please attach the output of SPSS.
2. Please attach some microscopic picture of the cell culture used.

Additional comments

no comment

Reviewer 2 ·

Basic reporting

1. Abstract is concise and has given the required gist of the article

2. Materials and methods:
- In sentence number 146: Add “KKU-100 and H69” cell lines as this apoptotic analysis by flow cytometry is done in all three cell lines

- In sentence number 154: Add "KKU-100" cell line as western blot analysis done in two cell lines.

Experimental design

1. Suitable experimental design in general

2. Good information regarding methodology to replicate the study.

Validity of the findings

Appropriate statistical tests are employed for this analysis.

The coverage of the discussion is well-written with adequate references.

Reviewer 3 ·

Basic reporting

PeerJ
#74668v1
Ethanolic extract of Ya-nang (Tiliacora triandra) leaf powder induces apoptosis in cholangiocarcinoma cell lines via induction of hyperacetylation and inhibition of growth signaling by Samankul et al.
In the present study, the author investigated the anticancer activity of Tiliacora triandra leaf powder ethanolic (TLPE) extract against cholangiocarcinoma cell lines. To accomplish this goal, They investigated the cytotoxicity of TLPE extract in CCA cell lines. They found that KKU-M213B cells were more sensitive to TLPE extract than KKU-100 cells. They showed that TLPE extract inhibited both human cholangiocarcinoma cell lines via apoptosis induction. Induction of cell cycle arrest was observed at G0/G1 phase only in KKU-100 cells. Apoptosis and cell cycle arrest inductions were confirmed by the levels of protein expression upon TLPE extract treatment. In the present study, the authors also analyzed the type of phenolic acids in TLPE extract and revealed 6 phenolic acids: 1) p-hydroxybenzoic acid, 2) vanillic acid, 3) syringic acid, 4) p-coumaric acid, 5) ferulic acid and 6) sinapinic acid.
In general, I’m supportive of this paper. I think it is very important to develop an alternative medicine for human cholangiocarcinoma treatment. Therefore, it is a very important area of research and in general, attracts a wide readership. In this manuscript, there is interesting and important data and I think at the end of the day it probably deserves to be published. With that said, I was disappointed in some of the technical shortcomings of this study. The author did not provide supportive data to validate the outcome of one experiment with other findings. English grammar needs extensive revision. I have made specific comments below, I do believe that a (significantly) revised manuscript would be acceptable for publication.

Abstract
Line 40: as chemotherapeutic …should be “as a chemotherapeutic:..

Introduction
Line 56: importantfor …should be…“important for”…

Line 58: Correct the sentence…“Current cholangiocarcinoma drugs include 5-flurouracil, gemcitabine and cisplatin (Valle et al., 2010), however, these drugs have a limitation in toxicity on the cells and expensive cost.
Line 65: Correct the sentence…“Tn Thailand and Laos, the leaves are extracted with water by rubbing with hands to produce a product called Nam Ya-Nang. T. triandra is used as a cooking ingredient and traditional medicine to treat illnesses such as fever and ma-laria, and also used as anti-pyretic drug, detoxication agent, anti- inflammation agent, anti-cancer agent, anti-bacterial agent, immune modulator and antioxidant.”

Materials & Methods
Line 177: “separating funnel”…..should be…. “a separating funnel.”

Result:
Line 209: “was”…..should be…. “were.”
Line 213: …proliferation… should be…. “the proliferation.”

Experimental design

Antiproliferative activity of TLPE extract against CCA cell lines:
I could not understand why the author used different concentrations for different time points for each cell line (Fig1A and 1B) and plotted graphs for cell viability separately (24hr vs 48and 72 hr). I would suggest that use same concentrations of drugs for all three-time points (24hr, 48 hr, and 72 hr) to compare the IC50 value at different time points.

TLPE extract induces cell cycle arrest in CCA cell lines:
These results are showing that the author did not synchronize the cells before the TLPE extract treatment. Asynchronous cells are in different phases before treatment. Therefore, cell accumulation is less and not consistent. I would suggest that the author should synchronize the cells by double thymidine block at the G0 phase and then treat them with different concentrations of TLPE extract for different time points.

TLPE extract modulates the expression of proteins involved in apoptosis and cell cycle arrest in CCA cell lines:
Western blot result showing that KKU-M213B cells (Fig 4A) inhibited the Bax expression upon TLPE extract treatment at 24 hr. However, this cell line has more apoptosis at 24 hr upon TLPE extract. According to apoptosis results, Bax expression should be high at a higher concentration of TLPE extract treatment as in KKU-100 cells. The author should detect the expression of the proteins involved in apoptosis-like cleaved PARP and Cleaved Caspase 3.

Validity of the findings

Antiproliferative activity of TLPE extract against CCA cell lines:
I could not understand why the author used different concentrations for different time points for each cell line (Fig1A and 1B) and plotted graphs for cell viability separately (24hr vs 48and 72 hr). I would suggest that use same concentrations of drugs for all three-time points (24hr, 48 hr, and 72 hr) to compare the IC50 value at different time points.

TLPE extract induces cell cycle arrest in CCA cell lines:
These results are showing that the author did not synchronize the cells before the TLPE extract treatment. Asynchronous cells are in different phases before treatment. Therefore, cell accumulation is less and not consistent. I would suggest that the author should synchronize the cells by double thymidine block at the G0 phase and then treat them with different concentrations of TLPE extract for different time points. It may be the possibility to arrest more cells after treatment.

TLPE extract modulates the expression of proteins involved in apoptosis and cell cycle arrest in CCA cell lines:
Western blot result showing that KKU-M213B cells (Fig 4A) inhibited the Bax expression upon TLPE extract treatment at 24 hr. However, this cell line has more apoptosis at 24 hr upon TLPE extract. According to apoptosis results, Bax expression should be high at a higher concentration of TLPE extract treatment as in KKU-100 cells.

Annotated reviews are not available for download in order to protect the identity of reviewers who chose to remain anonymous.

---

## Round 0.2 · Minor Revisions

Manuscript is significantly improved by the authors. However, there are still some minor concerns raised by the reviewer. Please address these concerns and resubmit accordingly.

Reviewer 2 ·

Basic reporting

This manuscript is revised
-English language is professional
-Relevant data/results provided to support Author’s arguments

Experimental design

-Material and methods are clearly explained compared to Initial manuscript version.

Validity of the findings

-Unnecessary information is removed in the revised manuscript.

Reviewer 3 ·

Basic reporting

PeerJ
#74668v2

The author claims that TLPE extract inhibited both human cholangiocarcinoma cell lines via apoptosis induction. Given the time that transpired between the first and second submissions, I was expecting a vastly improved manuscript. In the author's defense, some of the experimental data has indeed been improved by rearranging and explanation and is now more convincing. I have made some specific comments below.
Specific Comments:
Minor:
1. The paper in general is much better written than the original version and the authors are to be commended for making the effort to make the manuscript more grammatically correct and user-friendly. Nonetheless, there are still some places where there are typographical errors.
2. Rearranged Figure 5 and Figure 6 figure legends with the correct figure.

Experimental design

Combined Figure 5 with figure 4.

Validity of the findings

Figure 2: As per the author's claim, TLPE extract-treated KKU-M213B, KKU-100 and H69 cells significantly accumulate in the SubG1 phase of the cell cycle. While TLPE extract treatment in KKU-100 cells induced cell cycle arrest at G0/G1 phase. It is surprising that these arrested cells did not observe in the PI quadrant (Q1) of apoptosis (PI/annexin V FITC) data.
Figure 3: As the author mentioned in Figure legend; After 24 h- of TLPE extract treatment, cells labeled with Annexin V-FITC and Propidium iodide (PI) were analyzed by flow cytometry to determine the percentage of cells displaying an increase in early (panels Q4) and late (panels Q2) apoptosis. But I did not observe any increase in late (panels Q2) apoptosis after TLPE extract treatment. Although, cell-cycle arrest data shows a significant increase in the SubG1 phase of the cell cycle. It seems a technical mistake for gating the cells during data collection. As I believe, there should be some percentage of cells in the Q1 and Q2 panels if there is a significant increase in the sub-G1 population after TLPE extract treatment as shown in cell cycle analysis.

---

## Round 0.3 · accepted · Accept

Authors have adequately responded to all comments and manuscript now can be accepted in its current form.